# Microbes and Cancer: Friends or Faux?

**DOI:** 10.3390/ijms21093115

**Published:** 2020-04-28

**Authors:** Maria Manuel Azevedo, Cidália Pina-Vaz, Fátima Baltazar

**Affiliations:** 1Department of Microbiology, Faculty of Medicine, University of Porto, 4200-319 Porto, Portugal; 2CINTESIS, Faculty of Medicine, University of Porto, 4200-319 Porto, Portugal; 3Agrupamento de Escolas D. Maria II, 4760-067 V.N. Famalicão, Portugal; 4Life and Health Sciences Research Institute (ICVS), School of Medicine, University of Minho, Campus de Gualtar, 4710-057 Braga, Portugal; 5ICVS/3B’s—PT Government Associate Laboratory, 4710-057 Braga, Portugal; 6ICVS/3B’s—PT Government Associate Laboratory, 4835-258 Guimarães, Portugal

**Keywords:** cancer, infection, microbiome, cancer dysbiosis, tumor microbiome, cancer therapy response

## Abstract

Cancer is one of the most aggressive and deadly diseases in the world, representing the second leading cause of death. It is a multifactorial disease, in which genetic alterations play a key role, but several environmental factors also contribute to its development and progression. Infections induced by certain viruses, bacteria, fungi and parasites constitute risk factors for cancer, being chronic infection associated to the development of certain types of cancer. On the other hand, susceptibility to infectious diseases is higher in cancer patients. The state of the host immune system plays a crucial role in the susceptibility to both infection and cancer. Importantly, immunosuppressive cancer treatments increase the risk of infection, by decreasing the host defenses. Furthermore, alterations in the host microbiota is also a key factor in the susceptibility to develop cancer. More recently, the identification of a tumor microbiota, in which bacteria establish a symbiotic relationship with cancer cells, opened a new area of research. There is evidence demonstrating that the interaction between bacteria and cancer cells can modulate the anticancer drug response and toxicity. The present review focuses on the interaction between microbes and cancer, specifically aiming to: (1) review the main infectious agents associated with development of cancer and the role of microbiota in cancer susceptibility; (2) highlight the higher vulnerability of cancer patients to acquire infectious diseases; (3) document the relationship between cancer cells and tissue microbiota; (4) describe the role of intratumoral bacteria in the response and toxicity to cancer therapy.

## 1. Introduction

Cancer is one of the most prevalent diseases, representing the second leading cause of death worldwide [1,2], with approximately 18.1 million new cases and 9.6 million cancer related deaths estimated in 2018 [3]. Apart from this, it is associated with relevant social and economic burdens. In the present decade, cancer survivors in the United States increased from 13.8 to 18.1 million and the associated healthcare costs from $125 to $158 billion [4].

Cancer is considered not one disease but several diseases. The diversity of malignant tumors is substantial; with more than 250 clinico-pathological types and thousands of varieties of neoplasias described [5,6]. In addition, cells of the same tumor are generally morphologically, phenotypically and genetically heterogeneous. Furthermore, treatment can also induce cell diversification in metastases and recurrent lesions [5,6], rendering it very difficult to successfully eliminate malignant tumors. Cancer is also a multifactorial disease, resulting from a combination of genetic and environmental factors, which also contributes to tumor heterogeneity.

Infections induced by certain viruses, bacteria or fungi constitute risk factors for cancer development. Globally, 15% of cancers are a result of infection with oncogenic pathogens [7] and, in some cases, co-infection with different agents is known to increase the risk of cancer development [7]. Thus, the study of the mechanisms of infection-mediated cancer is of particular interest, aimed at both preventing cancer and improving current treatments. The main agents in terms of cancer burden are: *Helicobacter pylori* (5%), human papilloma viruses (HPV) (5%), hepatitis B (HBV) and C viruses (HCV), (5%), Epstein–Barr virus (EBV) (1%), and human immunodeficiency virus (HIV) plus human herpes virus (HSV) (1%) [8]. Cancer associated viruses can be acquired in utero, during infancy, early childhood or adolescence, but these agents have long latency periods before inducing carcinogenesis. On the other hand, the susceptibility to infectious disease is higher in cancer patients. That is—cancer is a double-edged sword. Chemotherapy is one of the most effective treatments for metastatic cancers [9]; however, some cancer therapies are also carcinogenic. Cancer therapy can change the host microbiota, increasing the susceptibility to infection, due to immunosuppression, and may increase the risk for cancer development. Finally, tumor microbioma has been associated with resistance to chemotherapy.

This manuscript focuses on:
(1)The relationship between infection and cancer;(2)The susceptibility of cancer patients to acquire infectious disease(s);(3)The role of the microbiota in cancer susceptibility;(4)The role of tumor microbioma in cancer therapy.

## 2. Infection and Cancer

The human organism contains more than 100 trillion microorganisms and these organisms play a significant role in human health and disease [10]. Nevertheless, only 10 species were identified by the International Agency for Cancer Research (IARC) as carcinogenic agents to humans [11]. These pathogenic microorganisms infect a great percentage of humans. However, most of these individuals do not develop cancer, since both the host characteristics and the microbial genotypes influence the susceptibility to develop cancer.

Human microbiome disruption is associated with different types of cancer, including gastric, colorectal, pancreatic and breast cancer. Moreover, the human body can be infected by innumerable environmental microorganisms, and cancers could be induced by bacteria, virus and fungi. In particular, human oncoviruses can drive carcinogenesis by integrating oncogenes into the host genome.

### 2.1. Viral Infections

According to recent publications, oncoviruses are responsible for nearly 12% of human cancers and are important factors in the activation of oncogenesis [12]. Oncoviruses play a major role in both the initiation and progression of cancer. DNA from certain oncogenic viruses can be integrated into the host genome, leading to the production of viral oncoproteins, which can subsequently inactivate tumor suppressor genes or activate oncogenes [12].

#### 2.1.1. Epstein–Barr Virus (EBV)

EBV was the first human virus directly implicated in carcinogenesis, and accounts for 1.8% of all cancer-related deaths worldwide [13]. EBV is essentially a B-lymphotropic agent, being associated with malignancies of B-cell origin. This virus is a main actor in the development of a wide range of cancers, both in immunocompetent and immunocompromised individuals. It is a ubiquitous gamma herpesvirus that persists for life and infects 90% of the population, in general without consequences in terms of health. Despite this fact, EBV is linked to several well-recognized malignancies, such as Burkitt’s lymphoma and nasopharyngeal carcinoma [14]. EBV transmission occurs via saliva and/or oral contact and genital secretions [15] (Figure 1).

Burkitt lymphoma is the greatest common form of non-Hodgkin Lymphoma in children and adolescents [16]. It is a highly aggressive and, often, life-threatening disease [17,18]. There is a geographical association with early EBV infection and Burkitt lymphoma pathogenesis (Figure 1). In the USA, 15–30% of the cases of sporadic Burkitt lymphoma are associated with EBV infections [19]. Importantly, the incidence of sporadic Burkitt lymphoma increases significantly when associated with HIV infections [20]. Carbone and co-workers reported that about 30%-50% of HIV-associated cases of Burkitt lymphoma are EBV positive [21]. In addition, lymphoproliferative Burkitt lymphoma disorder is reported to occur after organ transplant [22]. In light of this information, EBV monitoring may be important after transplant.

EBV is also associated with malignancies of epithelial origin, namely nasopharyngeal carcinoma (NPC). Nasopharyngeal carcinoma (NPC) is an infrequent disease and a unique squamous cell carcinoma of the head and neck [23] (Figure 1) (Table 1). In 2008, ~84,400 cases and 51,600 deaths from NPC were reported globally [24]; however, the geographic distribution is irregular. NPC is endemic in Southern Asia, and its etiology is strongly associated with EBV infection [25]. Some studies advocate that HPV may also be important in NPC progression [26].

Furthermore, EBV is associated with smooth muscle tumors. These are rare neoplasms that can occur in various anatomical locations, mainly in immunocompromised patients. The clinical signs are variable according to the size and organ affected [27]. The association of HBV with other varieties of cancer has been confirmed, including pancreatic [28], biliary tract [29] and breast cancer [30] (Figure 1) (Table 1).

#### 2.1.2. Human Papillomavirus (HPV)

According to the World Health Organization (WHO) [30,31], around 5% of cancers are instigated by HPV infections [8,32] (Figure 1). HPV infection is a global public health problem with great rates of transmission from mother to child, and it is highly associated with cervical, vaginal, vulvar, anal, rectal, penile and oropharyngeal cancers [33] (Table 1).

HPV infection is the most common sexually transmitted disease in the United States and Europe, with HPV16 and HPV18 being the higher-risk HPV types [33,34]. This infection is transmitted through genital or skin-to-skin contact and it is frequent in young women after the onset of sexual activity [35]. In light of this, the majority of the developed countries have implemented vaccination against HPV before initiation of sexual life. It is reported that 75% of sexually active woman will be infected by one or more types of HPV throughout their life. If infection persists, the viral oncoproteins can induce perturbation of the cell-cycle, resulting in cervical intraepithelial neoplasia. In the beginning, these lesions are generally no more than manifestations of HPV infection, but the risk of progression to cancer is higher if not detected and treated on time. Conversely, the evolution to cancer is slow, thus allowing the opportunity for detection. The incidence peak of cancer occurs around the age of 40 and can occur in 3–5% of women who acquire a high-risk HPV infection.

A data analysis from women displaying HPV positive test results suggests that the majority have limited knowledge about HPV. A survey conducted in the United Kingdom showed that only 2.5% of British women (n = 1620) identified HPV as the cause of cervical cancer [36]. Thus, several studies underline the importance of providing trustworthy information about HPV infection to women in general [37,38]. In light of this situation, health education on this subject should be a priority.

In 2012, a study by Torre and co-workers estimated that, globally, more than half a million of new cervical cancer cases emerged, and this disease was responsible for 266,000 deaths [39]. In the developing world, cervical cancer is the most commonly diagnosed cancer and the third leading cause of cancer-related death among females [39] (Figure 1). HPV infects keratinocytes in the mucosa, and persistent infection may lead to premalignant lesions and invasive cancer, especially cervical cancer. Nevertheless, screening for cervical cancer precursors by cytology has been very successful in developed countries, with mortality reductions of about 50% [40].

#### 2.1.3. Hepatitis Virus: B (HBV) and C Virus (HCV)

Both HBV and HCV are transmitted through exposure to body fluids of infected individuals. HBV infection is considered a global public health problem, due to the high rates of transmission from mother to child [41]. Chronic HBV infection is associated with hepatocellular cancer (HCC) (Figure 1). It has been described that 4% of untreated patients with HBV infection develop HCC [42]. About 80% of the HCC cases are linked to chronic infections with either HBV or HCV, and both viruses have been classified by the IARC as human carcinogens. It is estimated that HBV and HCV cause 50–55% and 25–30% of all HCC, respectively [43]. However, the mechanisms of carcinogenesis and/or cancer progression remain to be fully elucidated for both viruses [44,45].

More recently, HBV has been linked to other kinds of cancers such as non-Hodgkin lymphoma [46], pancreatic cancer [28], biliary tract cancer [29] and breast cancer [30] (Figure 1) (Table 1). Likewise, HCV infection constitutes a risk factor, not only for HCC but, also, for non-Hodgkin lymphoma and B-cell non-Hodgkin lymphoma [47,48]. Other studies have shown a possible association between HCV infection and both thyroid [49] and liver cancer [50] (Table 1).

#### 2.1.4. Human Immunodeficiency Virus (HIV)

Human immunodeficiency virus (HIV) infection is linked with acquired immune deficiency syndrome (AIDS). Currently available epidemiologic and phylogenetic analyses imply that HIV was introduced into the human population around 1920 to 1940. HIV is classified into type 1 and 2 (HIV-1, HIV-2), with evidence showing that HIV-1 evolved from non-human primate immunodeficiency viruses from Central African chimpanzees (SIVcpz) and HIV-2 from West African sooty mangabeys (SIVsm) [51,52,53].

Kaposi sarcoma (KS) is the most frequent neoplasm among patients living with HIV infection [54] (Figure 1) (Table 1). KSHV, also called herpesvirus-8, was discovered in 1994, with its prevalence differing deeply among populations. Some types of KS are related with immune suppression [55]. For instance, HIV is considered a crucial cofactor in KS pathogenesis, as it was demonstrated that the employment of antiretroviral therapy reduces KS incidence [56]. The presentation of this disease varies from cutaneous limited lesions to fulminant disease involving several organs. It is noteworthy that patients infected with HIV are also at higher risk of developing secondary cancers due to chronic immunosuppression. Additionally, the risk of anal, liver, lung, oropharyngeal cancers, and NHL and Hodgkin’s lymphoma (HL) is remarkably higher in HIV-infected patients compared to the general population [57], as shown in studies from the United States (Figure 1). Reporting to the time period between 1981–1995, an important database indicates that patients infected with both HIV infection and KS positive (HIV-KS patients) were at a 198-fold higher risk of developing non-Hodgkin’s lymphoma (NHL), compared to the general population of the United States [58] (Figure 1).

Not only HIV positive individuals can present KS; immunocompromised people infected with KS-associated herpesvirus (KSHV), aging people, children in KSHV-endemic areas and patients submitted to transplants are also at risk [59].

## 3. Bacterial Infections

### 3.1. Helicobacter pylori (H. pylori)

*H. pylori* infects more than 50% of the world’s population [60], rendering it the most frequent infection in the world [61]. It is the most central factor of gastric cancer (GC) risk (Figure 1) (Table 1); due to its characteristics, *H. pylori* is able to adapt to the extreme acidic conditions of the stomach, to establish infection and disturb host mucosa homeostasis, resulting in gastric pathogenesis and ultimately cancer [61].

The frequency of infected people may somehow be related to race. The prevalence is about 60% of Hispanics, 54% of African Americans and 20–29% of White Americans. *H. pylori* infections are typically developed during childhood and usually remain latent for decades. This bacterium stimulates the production of interleukins 1β (IL-1β), 2 (IL-2), 10 (IL-10), 12 (IL-12), interferon (IFN) and tumour necrosis factor (TNF) as the gastric immune response. The inflammation induced by *H. pylori* infection leads to high gastric endothelial cell turnover. Deregulation, particularly in association with more virulent strains of *H. pylori*, can trigger GC. *H. pylori* and can also induce atrophy of the gastric mucosa and impaired acid secretion. The atrophy of the gastric mucosa can lead to molecular and phenotypic changes causing cancer [62]. A study conducted in California revealed that 3% of patients with gastric intestinal metaplasia (GIM) developed gastric cancer in a period of 4.6 years after diagnosis. Discrepant results have been published concerning the link between *H. pylori* and other gastrointestinal cancers, including pancreatic cancer [63].

Epidemiological studies confirm that the eradication of *H. pylori* prevents gastric carcinogenesis and mild gastritis [64]. Current studies also demonstrate that combined primary (*H. pylori* eradication), and secondary (mainly endoscopy) measures may prevent, or limit, the development of gastric oncogenesis [62]. The effectiveness of gastroprotective drugs has also been shown in the prevention of gastric cancer in patients, with or without *H. pylori* infection [65]. The development of GC constitutes a multifaceted cascade of events, characterized by interactions between *H. pylori*, host and environmental factors [66]. Gastric microbiome can constitute a marker of host health, and influences the inflammatory response of upper gastrointestinal cancers [67].

It is important to reinforce that, despite all the advances in the surgical techniques and chemotherapy, there is still no cure for GC [68]. Gastric cancer occurrence and mortality remain extremely high in East Asia, Latin America and Eastern Europe, and within specific groups in the USA [69,70]. It represents the fourth leading cause of cancer deaths worldwide [67], with a 5-year overall survival rate at less than 30% [62]. In 2018, data registries from 185 countries worldwide indicated about 1 million new cases of GC (5,7% of all cancers) and 800,000 deaths (8,2% of all cancers) [3].

It is important to highlight that sanitation, hygiene, clean water provision, and improvements in food preservation, variety, and availability, contributed significantly to the decrease in GC incidence [69,70]. In terms of education, literature reviews highlight the lack of studies evaluating public awareness of *H. pylori* infection, particularly among populations at an increased risk for GC [71]. Thus, it is crucial to educate the population for the risk factors related with this type of infection.

### 3.2. Fungal Infections

Several publications link the presence of fungal infections with a higher risk of developing cancer. Some strains of filamentous fungi, including *Aspergillus flavus*, *Aspergillus parasiticus* and *Aspergiluls nomiusm*, produce aflatoxins able to induce liver cancer [72] (Table 1). According to Sohrabi and collaborators, the tumor burden increases significantly after infection with *Aspergillus* [73] (Figure 1).

Other studies indicate that yeast infections are also associated with a higher risk of cancer and are able to promote cancer progression; such is the case of opportunistic *Candida albicans* infections [74,75]. Specifically, the role of Candida infections has been demonstrated in lung cancer and it has been documented as a risk factor for development of oral carcinoma [76] (Figure 1). There are several mechanisms pointed out by which *C. albicans* can promote cancer, such as (i) production of carcinogens like acetaldehyde, (ii) induction of inflammation, (iii) molecular mimicry and (iv) Th17 response [77].

### 3.3. Helminth Infections

Helminths include free-living and parasitic Platyhelminthes and Nematoda, which infect millions of people worldwide. Some Platyhelminth species of blood (*Schistosoma haematobium, Schistosoma japonicum, and Schistosoma mansoni*) and liver (*Clonorchis sinensis* and *Opisthorchis viverrini*) flukes are known to be involved in human cancers (Figure 1). Metagenomic studies with microbiomes have been undertaken, revealing that parasitic infections lead to inflammation and disease development, which results from the interaction of helminth with the host, potentially inducing such processes [78,79,80,81,82]. Phylogenetics and molecular evolution studies may contribute to identify Helminth and human biomarkers in different types of cancers, further providing a robust framework for the functional prediction and rational design of their experimental characterization. These approaches may contribute to improve personalized treatment to avoid cancer progression and drug resistance.

### 3.4. Schistosoma haematobium (S. haematobium)

Schistosomiasis is endemic in 78 countries [83], and infection with *S. haematobium* reflects the high burden of the disease in parts of Northern and sub-Saharan Africa [8]. Epidemiological studies suggest the association between *S. haematobium* infection and the onset of bladder cancer [84] (Figure 1) (Table 1). In Western countries, the peak incidence of bladder cancer occurs in the sixth or seventh decade of life, and only 12% of cases occur in people under 50 years of age [85,86].

Parasite eggs are deposited in the bladder, which causes an intense inflammatory reaction. These inflammatory conditions could promote premalignant lesions, which could be followed by malignant transformation of the urothelium [87,88]. Exposure to the parasite can result from the fecal or urinary contamination of freshwater which comes into contact with the human skin. Accordingly, populations with adequate sanitation have significantly lower levels of *Schistosoma* infections [89]. A recent systematic review and meta-analysis demonstrated the relationship between adequate sanitation and good hygiene with lower schistosomiasis infection [90]. Additionally, the use of soap when bathing may also protect from schistosome infection [91,92].

Botelho and co-workers have shown that exposure to the total antigen of *S. haematobium* increased the proliferation rate of normal urothelial cells. Additionally, intravesical administration of the same antigen in murine models leads to the development of dysplastic lesions, suggesting that this antigen may be associated with the cancerization process [93,94]. The degree of intensity of this infection plays an important role in the induction of different types of carcinoma of the bladder. Squamous cell carcinoma is generally associated with a high to moderate parasitic load, whereas urothelial carcinoma occurs more frequently in areas associated with lower rates of infection.

## 4. Susceptibility of Cancer Patients to Acquire Infectious Diseases

### 4.1. Bacterial Infections

The functions of the immune system are organized in order to maintain a balance with the microbiota and to fight against most microbial invasions. In contrast, in immunocompromised patients induced by immunosuppressive treatments, including cancer chemotherapy, these functions are impaired and individuals are at an increased risk of infection. Chemotherapy is one of the most effective treatments for metastatic cancers [9]; however, these cancer therapies change the interactions between the host and microbiota. In patients with solid tumors, the usual infection sites include the skin and skin structures (including surgical site infections), the bloodstream (including infections associated with central venous catheters), the lungs, the hepato-biliary and intestinal tracts, and the urinary tract [95].

Tumor metastases leading to vessel obstruction and surgical procedures with disruption of anatomic barriers may be implicated in cancer patient infections. Cases of pneumonia in children have been reported to be associated with pulmonary metastases [96]. This kind of infection usually responds poorly to antibiotic therapy. Biliary tract obstruction secondary to cancer can also result in ascending cholangitis [97]. Similarly, urinary tract infections are frequent in patients with bladder or prostatic tumors. Additionally, patients with cancer develop bacterial infections frequently, especially, but not exclusively, during episodes of neutropenia [98].

Cancer is also associated with *Mycobacterium tuberculosis* and other mycobacterial infection. Patients with cancer constitute a group at increased risk to develop tuberculosis [99]. Particularly, hematologic cancer patients present the maximum rates of active tuberculosis, followed by head and neck cancers, lung cancer, and breast cancer. This fact is likely related to intrinsic immunosuppression induced by the cancer itself, the immunosuppressive effects of chemotherapy, or other host factors that may increase the susceptibility to both cancer and tuberculosis.

### 4.2. Fungal Infections

Cancer individuals treated with chemotherapy and other immunocompromised patients, such as after hematopoietic stem cell/organ transplantation or AIDS patients, have an increased risk of acquiring invasive fungal infections (IFI). Approximately 1.2 billion individuals worldwide suffer from fungal infections, and the occurrence of these infections has significantly increased in recent years, due to a rise in the number of immunocompromised patients [100,101,102]. IFI are a major cause of morbidity and mortality in neutropenic cancer patients, in those with acute leukemia and in hematopoietic stem-cell transplantations. Although uncommon pathogens, including non-fumigatus *Aspergillus* species, *Zygomycetes* and *Fusarium*, have emerged, most IFI are caused by *Candida* species and *Aspergillus fumigatus* [103,104]. *Candida* spp. constitutes the most common agent related to both fungal and fungal biofilm infections [105]. The prevalence of candidiasis in hospitals has increased due the use of catheters and immunosuppressive treatments, such as chemotherapy [106,107]. In cancer patients, candidemia may vary according to the type of cancer [108]. On the other hand, Ramirez-Garcia and collaborators reveal that *C. albicans* infection increases the risk of carcinogenesis and metastasis [77].

Concerning *Aspergillus*, the most frequent species involved in IFI in neutropenic patients are *A. fumigatus*, *A. terreus* and *A. flavus*, however new species are emerging [109,110]. The incidence of invasive aspergillosis infections may be as high as 10%–20% in patients undergoing gallogeneic haematopoietic stem-cell transplantations [111]. Invasive pulmonary aspergillosis, caused by several *Aspergillus* species, is an important cause of morbidity and mortality in cancer, transplanted, and other patients with immunodeficiency [112,113,114]. Mortality rates with these infections range from 20% to 47% despite prophylaxis with use of new antifungal agents active against molds [115]. Moreover, the clinical diagnosis of these infections is challenging, often being misdiagnosed as bacterial infections, which can have fatal consequences [116,117]. Infection with *Fusarium* species are relatively rare, but significant in immunocompromised patients. Disseminated infection with *Fusarium* is seen in patients with prolonged neutropenia after chemotherapy. The prognosis of disseminated disease with neutropenia is very poor, with almost 100% mortality [118].

Mucormycoses, induced by Mucorales, are significant IFI developed in patients with hematologic malignancies and among host recipients of solid organ transplants [119]. This infection is also associated with considerable mortality rates.

It has been advocated that early treatment of IFIs is a crucial factor in improving health outcomes; however, antifungal treatment is frequently delayed, due to diagnostic challenges related with the non-specific symptoms of IFIs. Moreover, there are several apprehensions with regard to prophylaxis, including the high costs, toxicity and emergence of drug resistance [120].

## 5. Tissue Microbioma and Cancer Susceptibility

Studies on the role of microbiota in cancer are at their infancy, but data show that microbiota may influence carcinogenesis and response to cancer therapy. A large quantity of studies on human microbiome demonstrate that the microbiota varies from healthy to diseased persons. An altered microbiome can lead to gene overexpression, which may be associated with complex diseases including cancer. In the human body, the microbiota of each organ varies among individuals, and their effect on carcinogenesis is also distinctive in each organ.

According to Garrett [121], microbiota could contribute to carcinogenesis by altering the balance between the host cell proliferation and death (by changing the immune system and influencing metabolism). Mucosal surface barriers permit host-microbial symbiosis [122], however cancer could arise when mucosal surface barriers are disrupted and microorganisms and immune systems are present in places and assemblages for which they have not coevolved. After rupture of these barriers, microbes can modulate the immune responses in evolving tumor microenvironment by eliciting proinflammatory or immunosuppressive programs [121]. The resulting production of reactive oxygen and nitrogen species, cytokines, and chemokines can contribute to tumor growth and spread. On the other hand, microorganisms participate in host metabolic activities, and their metabolites can induce inflammatory processes, interfering in the balance of tissue cell proliferation and death [123]. The arising of cancer in some locations could be induced by microbial dissimilarities, such as the greater vulnerability to cancer in the large intestine due to the higher microorganism density when compared to the small intestine [124]. Thus, the microbiota of each organ of the human body is different, and their effect on inflammation and carcinogenesis are also distinct in each organ.

Dysbiosis as a result of cancer-induced immunodeficiency, chemotherapy treatment regimens or antibiotic use, could also raise the risk of bloodstream and *Clostridium difficile* infections, by disrupting the gut microbiome’s ability to resist pathogen colonization or by weakening the intestinal barrier. Tai and co-workers described that the frequency rate of *Clostridium difficile* infections in hospitalized oncologic children was over 15 times higher compared to those without cancer [125]. Another study demonstrated that pediatric oncology patients with acute lymphoblastic leukemia, under several circles of therapy, showed reduced microbial diversity [126]. Wang and co-workers also found reduced diversity and abundance of oral microbiome in patients with acute lymphoblastic leukemia [127].

Microbiota uses crucial mechanisms such as inflammation, metabolism and genotoxicity to modulate carcinogenesis [128]. Current studies also demonstrate that the gut microbiome may affect the response to cancer therapy, by modulating the inflammatory response of the host. The metabolic activity of microorganisms could activate, disable or increase, treatment toxicity. A study from 1993 reported the capacity of the microbiome to interfere with medicines in a group of Japanese who developed *Herpes zoster* while suffering from cancer. *Herpes zoster* medication was transformed by the normal digestive microbiota into a component that made the medication to treat cancer lethally toxic. In this line of evidence, Karin and co-workers stated that an intact microbiome is needed for the successful control of tumor progression [129].

### 5.1. Breast Cancer

As stated above, cancer can be associated with the specific microbial environment in the tissue of origin. A recent study reported that the microbiome found in breast tissue varies between women with and without breast cancer [130] (Figure 2). The researchers detected higher levels of *Bacillus, Enterobacteriaceae, Staphylococcus, Comamonadaceae* and *Bacterioidetes* in women cancer samples and, in contrast, healthy breast tissue revealed higher levels of *Prevotella*, *Lactococcus*, *Streptococcus Corynebacterium* and *Micrococcus* species [130]. A study conducted by Hieken and collaborators [131] detected high levels of phyla Firmicutes, Actinobacteria, Bacteroidetes, and Proteobacteria in healthy breast tissues, compared to breast cancer tissues where a lower abundance of *Fusobacterium, Atopobium, Gluconacetobacter, Hydrogenophaga and Lactobacillus* genera were found. Another study on breast cancer also showed distinct microbial communities between normal and cancer breast tissues. The malignancy is associated with enrichment in the genera *Fusobacterium*, *Atopobium*, *Gluconacetobacter*, *Hydrogenophaga* and *Lactobacillus* [131].

This fact raises an important issue: can women modulate their breast microbiome in order to prevent high levels of *E. coli* or *S. aureus* colonization? Some studies revealed that drinking fermented products, such as kefir, is associated with lower risk of breast cancer. Other studies using animal models showed that orally ingested *Lactobacillus* could have a protective role against breast cancer development [130]. In light of this, upcoming studies are needed to demonstrate the role of probiotics as a preventive measure against breast cancer.

Furthermore, it has been emphasized that estrobolome (hormone-related microbiome) bacterial composition possibly contributes to the development of hormone-driven malignancies, such as breast cancer [132]. However, this depends on host factors and associated behaviors, such as age, ethnicity, diet, alcohol consumption and antibiotic use, which may exert selective pressures on bacterial populations [133,134,135]. In this context, it was revealed that some antibiotics such as clarithromycin, metronidazole and ciprofloxacin induce a reduction in the biodiversity/abundance of some bacterial communities disturbing the equilibrium of the gut microbiome [136,137], which constitutes a higher risk of breast cancer [138]. Another study developed by Goedert and co-workers detected a link between fecal microbiome lower diversity in postmenopausal women and breast cancer. This study showed that 87% of the woman had ER-positive tumors when compared to healthy control women [139].

### 5.2. Digestive Tract Cancer

#### 5.2.1. Gastric Cancer

Gastric carcinoma is a main global health problem, with over 1 million new cases each year [3]. Gastric chronic infection with *H. pylori* induces reduced acid secretion, which may stimulate the proliferation of different bacteria in the gastric mucosa. This modification in the gastric microbiome may induce damage to the gastric mucosa and contribute to malignancy. Recently, Ferreira and co-workers showed that there are significant differences between the gastric microbiota of patients with gastric carcinoma and chronic gastritis. Dysbiosis in gastric carcinoma was characterized by decreased microbial diversity, reduced *H. pylori* burden and enrichment of other bacterial genera, consistent with a genotoxic microbial community [140] (Figure 2). These results are in accordance with a previous report, showing lower bacterial diversity in patients with gastric cancer compared with patients with non-atrophic gastritis [141].

Proteobacteria, Firmicutes, Bacteroidetes, Actinobacteria and Fusobacteria are the leading phyla present in the gastric mucosa [142,143,144]. On the other hand, in gastric carcinoma, there is a significant decrease in *Helicobacter* burden, and an increased abundance in several taxa such as Citrobacter, Clostridium, Lactobacillus, Achromobacter and Rhodococcus. These taxa exist usually in the intestinal mucosa as commensals, but they can become opportunistic pathogens [145,146]. Another study described that *Lactobacillus* was one of the main genera detected in Swedish patients with gastric cancer [147].

#### 5.2.2. Colorectal Cancer

Globally colon and rectum cancer (CRC) is responsible for about 500,000 and 30,000 deaths per year [3]. especially CRC affects the large intestine where a high bacterial load is prevalent [148]—the human intestinal tract is a habitat of about 1000 bacterial species [149]. Investigations have revealed a relationship between gut microbiota modification and colorectal cancer, as depicted in (Figure 2). A very recent study showed that intestinal bacteria play an important role in CRC development [150]. In healthy individuals the leading groups of bacteria are Firmicutes, Bacteroidetes and Actinobacteria, while Proteobacteria and Verrucomicrobia are present in minor quantities [151]. Conversely, intestinal bacteria in CRC patients seem to display a different profile. Current literature provides ample support for the contribution of the composition of intestinal bacteria in cancer development [152,153,154,155].

Lucas and co-workers show evidence for the key role of the gut microbiota in colorectal cancer progression [156]. The gut microbiota builds a symbiotic relationship with the host, and this is implicated in metabolic, immunological and protective functions in individual health. Data published in 1995 identified 15 bacterial species linked with higher risk to develop CRC (2 *Bacteroides* species, 2 *Bifidobacterium* species, 5 *Eubacterium* species, 3 *Ruminococcus* species, *Streptococcus hansenii*, *Fusobacterium prausnitzii* and *Peptostreptococcus* products) [157]. A study performed by Chen and co-workers, with 46 patients with CRC and 56 healthy subjects, involving analyses of the microbiota composition of different intestinal compartments, indicated that the mucosa-associated bacterial composition was expressively different in healthy and CRC patients. The microbiota of CRC exhibited lower diversity comparatively to the healthy tissues. The mucosa of the CRC patients is enriched in *Fusobacterium, Porphyromonas*, *Peptostreptococcus*, *Gemella*, *Mogibacterium* and *Klebsiella*, but are reduced in *Feacalibacterium*, *Blautia*, *Lachnospira*, *Bifidobacterium* and *Anaerostipes* [158]. Recent research showed that in CRC patients, the principal phylum is Firmicutes, and in healthy subjects, is Proteobacteria [159]. The mechanisms by which bacteria can induce colorectal cancer are diverse, including: (i) enhanced release of toxins produced by bacteria; (ii) reduction of bacterial benefits in terms of bacterial-derived metabolites; (iii) disruption of epithelial barrier; (iv) production of pro-carcinogenic compounds and, (v) modifications in the intestinal microbiota. The mentioned mechanisms induce chronic inflammation and increase cellular proliferation, favoring CRC progress [128].

#### 5.2.3. Pancreatic Cancer

Pancreatic cancer is one of the most lethal cancers worldwide [160], with several studies suggesting a possible role of some microorganisms in the pathogenesis of this kind of cancer (such as *H. pylori* [161]). It is, in fact, a recognized inflammation-driven cancer, with large preclinical and clinical evidence indicating that bacteria likely influence this process, since they activate immune receptors, perpetuating cancer-associated inflammation.

A recent meta-analysis highlights an increased risk associated with the development of pancreatic cancer for cag-A-negative *H. pylori* strain [162], while a study by Farrell et al. [163] presented convincing evidence by revealing associations between specific profiles of oral bacteria and an increased risk of pancreatitis and pancreatic cancer. A recent review by Archibugi and co-workers supports that several risk aspects related with pancreatic cancer are linked to microbiome modification [164], reporting a connection between oral microbiome associated with periodontitis and a higher risk of pancreatic cancer. Microbial diversity modifications, involving bacterial species such as *Porphyromonas, Actinomycetes*, *Neisseria, Streptococcus, Bacteroides, Bifidobacteria* and *Fusobacterium* have been associated with pancreatic cancer [163,165,166,167,168,169,170,171] (Figure 2). Finally, more recently, McAllister and collaborators demonstrated that intra-tumoral bacteria and fungi are implicated in in pancreatic cancer carcinogenesis [172].

#### 5.2.4. Oral Cancer

Several bacterial species have been associated with oral cancer [173]. More recently, investigations have been carried out to identify changes in bacterial microbiota in oral squamous cell carcinoma (OSCC) (Figure 2). High levels of *Porphyromonas*, *Prevotella* and *Fusobacterium* species were reported in patients with OSCC lesions. Mager et al. described the occurrence of *F. periodonticum* in saliva of patients with OSCC [174]. *Capnocytophaga gingivalis*, *P. melaninogenica* and *S. mitis* were also highly abundant in saliva of patients with OSCC [174,175]. Large amounts of *Aggregatibacter*, *Lautropia*, *Haemophillus*, *Neisseria*, and *Leptotrichia* were significantly higher in the control. Another study found low levels of *Streptococcus sp* and *Rothia sp* in OSCC patients, despite high levels of *Bacteroidetes* and *Fusobacteria* [176]. Beyond OSCC, changes in oral microbiome are also connected with oral premalignant conditions [177]. Malignant transformation appears to result from the formation of by-products of microbial metabolism. Examples of products with genotoxic activity are acetaldehyde and N-nitrosamine compounds present in the saliva of smokers and alcohol drinkers [178].

## 6. Role of Microbioma in Cancer Drug Response and Toxicity

Advances in cancer therapy have been significant in recent years, however resistance to treatment and associated toxicity remains a notable challenge. This part of the text will explore the role of bacteria in the response to cancer therapy, including microorganism-modulated therapy efficacy and toxicity.

According to the literature, the gut microbiota can influence the efficacy of anti-cancer chemotherapy. The efficacy of oxaliplatin, a platinum-based chemotherapy agent, has been described to be influenced by the microbiota together with the immune system [179]. The gut microbiota prime myeloid cells for production of high-levels of reactive oxygen species (ROS). In this way, the increase oxidative stress production of ROS enhances; consequently, this augments oxaliplatin-associated DNA damage, triggering cancer cell death [179]. On the other hand, the use of cyclophosphamide, a chemotherapeutic alkylating agent used in hematologic malignancies and solid tumors, leads to injury to the small intestinal epithelium. The subsequent barrier breach results in gut microbiota-dependent, T helper (TH) cell–mediated antitumor responses [180]. Additionally, the anti-tumoral efficacy of pyrimidine nucleoside analogues (e.g., 5-fluorouracil, 5-FU) can be compromised by the metabolic activity of intratumoral bacteria. It was demonstrated in vitro that the anti-tumoral efficacy of these pyrimidine analogues decreased in cells infected with *Mycoplasma hyorhinis*, as a result of bacterial degradation of the anticancer drugs, through the activity of thymidine phosphorylase [181]. Another study also showed that intratumoral bacteria can metabolize the chemotherapeutic drug gemcitabine into an inactive form [182]. Gemcitabine metabolism is dependent on the expression of a long isoform of the bacterial enzyme cytidine deaminase (CDD_1_), seen primarily in Gammaproteobacteria. In a colon cancer mouse model, the authors demonstrated that gemcitabine resistance was induced by the presence of intratumor Gammaproteobacteria, dependent on bacterial CDD_1_ expression, which was abrogated by co-treatment with ciprofloxacin (quinolone antibiotic). This study suggests that intratumoral bacteria can compromise the efficacy of this drug treatment of malignant tumors [182]. These results were supported by an analysis of 113 patients with pancreatic ductal adenocarcinoma, which revealed that 76% were positive for bacteria, mainly Gammaproteobacteria. Thus, these studies suggest that microbiota should be considered as an important factor in the response to chemotherapy, and antitumoral drug responses can be potentiated by the co-administration of antibiotics, but such mixtures must be the subject of further exploration in preclinical and clinical scenarios.

Importantly, there is evidence for modulation of immunotherapy efficacy by gut microbiota. In a melanoma murine model, Paulos et al. [183] demonstrated that whole body mice irradiation improved the efficacy of CD8+ T cell adoptive transfer therapy. Irradiation induced a bacterial lipopolysaccharide (LPS) release, which activated an innate immune response, boosting the anti-tumor activity of CD8+ T cells. On the contrary, antibiotic treatment or LPS neutralization decreased immunotherapy anti-tumor response. In colon carcinoma and melanoma murine cancer models, Iida et al. showed that mice antibiotic treatment compromised the efficacy of immunotherapy using anti-IL-10/CpG oligodeoxynucleotides (ODN). This lack of effect resulted from a decrease in the gut microbiota, and involved a decline in the production of pro-inflammatory cytokines by tumor-associated monocytes. Additionally, the impact of gut microbiota on the efficacy and toxicity of the more recent immune checkpoint inhibitor (ICI) therapy has been demonstrated. On study showed that germ-free or antibiotic-treated mice did not respond well to cytotoxic T-lymphocyte antigen 4 (CTLA-4) antibody inhibition [184]. Efficacy was restored by feeding mice orally with Bacteroides spp, and was associated with the increase in intratumoral mature dendritic cells (DCs) and the elevation of anti-tumor Th1 response. This association of anti-CTLA-4 response and gut microbiota was subsequently demonstrated in clinical studies [185]. Furthermore, response to anti-programmed cell death 1 (PD-1)/PD-L1 therapy could be predicted by the microbiota profile of patients with solid epithelial tumors. The presence of *Faecalibacterium* sp. in the gut microbiome of melanoma patients was associated with anti-PD-L1 response, while *Bacteroides thetaiotaomicron, Escherichia coli, and Anaerotruncus colihominis* were associated with a lack of response [186]. These results encourage the use of microbial targeting to boost the efficacy of anti-tumor immunotherapy, however, most preclinical studies are performed in immunocompromised murine models and clinical translation becomes limited.

Besides cancer pharmacotherapy, there is evidence for the role of gut microbiota composition in the response to other anti-cancer treatment modalities, namely radiotherapy and surgery. Evidence is still weak for radiotherapy, since there are only a few published studies supporting this association [187]. One of the studies is by Cui et al. [188], who reported an association between the composition of intestinal microbiota and radiosensitivity, in which antibiotic-treated mice presented higher survival rates after irradiation, compared to antibiotic-untreated mice. Evidence for the potential impact of gut microbiota in surgery is limited to CRC. The existing studies support the role of the gut microbiota in the pathogenesis of anastomotic leaks, common life-threatening postoperative complications. Several studies report that the use of oral antibiotics in colorectal surgery protect patients from these postoperative complications, however the mechanisms are still poorly understood [189].

From the above described, it appears that there is potential in the modulation of the gut microbiome as a rational approach to enhance the efficacy of cancer therapy. Indeed, several studies support the use of selective antibiotics (some examples described above), but also fecal transplantation, as well as probiotics and prebiotics (for a review, see [190]). However, most of the time, the evidence is not strong enough and the mechanisms behind their antitumor activity are still unclear, which demands further studies.

Despite the importance and proven efficacy of the chemotherapeutic strategies, they also induce severe toxicity to normal cells, especially highly proliferative tissues. As an example, several cancers are treated with irinotecan (topoisomerase-1 inhibitor) in combination with other agents, and a usual side effect is the onset of diarrhea. In some cases, this situation leads to hospitalization. Within the intestinal lumen, microbial-produced β-glucuronidases regulate the levels of irinotecan’s bioactive form and thus influence irinotecan activity and toxicity [191]. Thus, the use of oral bacterial β-glucuronidase inhibitors blunt the dose-limiting toxicities of irinotecan, not damaging host cells or destroying bacteria. These data suggest that microbial metabolism modulation constitutes a rational target in cancer care [191]. Another example is the involvement of *Bacteroides spp*, members of intestinal microbiota, in the toxicity of 5-FU/sorivudine therapy. Bacteroides species induce sorivudine conversion to the intermediate BVU which, in turn, inhibits degradation of 5-FU, raising 5-FU plasma concentrations with consequent higher toxicity [192]. The composition of gut microbiota may also influence the toxicity induced by radiotherapy. A clinical study by Ferreira et al. [193], reported an association between gut microbiota composition and radiation-induced enteropathy, in which higher loads of *Clostridium, Roseburia and Phascolarctobacterium* appeared increased in patients with enteropathy.

## 7. Conclusions

As stated above, it is clearly established that infection with several microorganisms constitute a cause of cancer. Important examples are HPV infection in cervical cancer, *H. pylori* in gastric cancer, *C. albicans* in oral squamous cell carcinoma, and *S. haematobium* in bladder cancer. On the other hand, microorganisms, especially bacteria, are an important part of the human organism ecosystem, which is known as microbiota. In general, the relationship established between the host and microbiota is symbiotic, with important functions to the human organism, including the production of nutrients, vitamins and defense against pathogens. However, alterations in the human microbiota can lead to infection that can play a role in carcinogenesis and the susceptibility to acquire cancer can also be influenced by the microbial composition of the human body. Additionally, particularities of the cancer type, or the treatment regimens, can make cancer patients more vulnerable to the acquisition of infectious diseases. Lastly, but no less important, microbiota and/or intratumoral bacteria can influence the efficacy and toxicity of cancer therapy. Thus, modulation of microbial activity may help in the antitumor treatment response.

On the whole, it seems to be clear that microbial communities that colonize the human body play a crucial role in health and disease. In healthy conditions, microbes are allies and live in dynamic equilibrium with the host. However, in disease conditions, a microbial imbalance is created and they may become faux, having the capacity to sustain disease, including cancer. Thus, “fine tuning” the human microbiome, taking advantage of the relationship with the microorganisms that colonize our body, would likely be a strategy in fighting disease in a more effective way. In order to master this approach, further investment in research directed to the different aspects of our interaction with the microbes is mandatory.

## Figures and Tables

**Figure 1 ijms-21-03115-f001:**
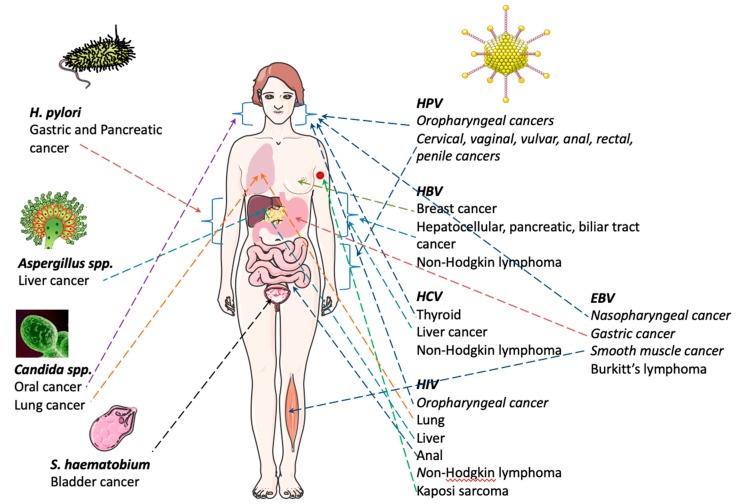
Infectious agents and associated cancer types. This figure was produced, in part, using Servier Medical Art (https://smart.servier.com).

**Figure 2 ijms-21-03115-f002:**
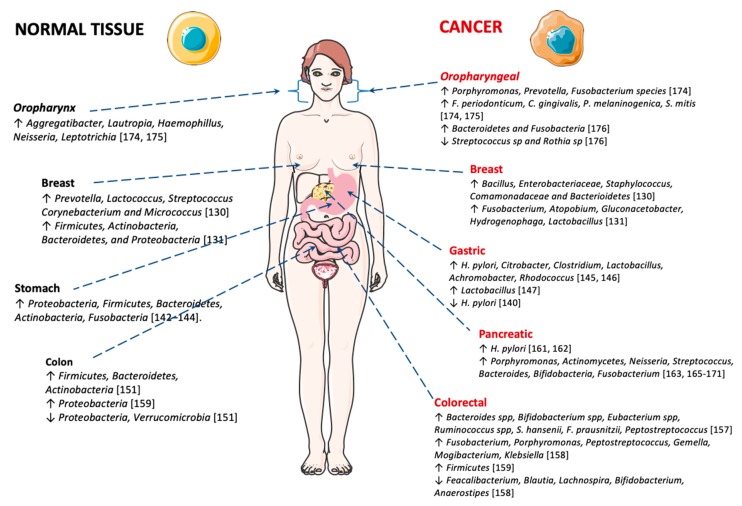
Microbiota in normal and cancer tissues. This figure was produced, in part, using Servier Medical Art (https://smart.servier.com).

**Table 1 ijms-21-03115-t001:** Worldwide number of new cases and deaths for cancers with associated infectious agents identified (Cancer Facts & Figures 2019).

Cancer Types	No. of New Cases (% of All Sites)	No. of Deaths (% of All Sites)
Bladder	549,393 (3)	199,922 (2,1)
Breast	2,088,849 (11,6)	626,679 (6,6)
Cervical	569,847 (3,2)	311,365 (3,3)
Colon	1,096,601 (6,1)	551,269 (5,8)
Gastric	1,033,701 (5,7)	782,685 (8,2)
Kaposi	41,799 (0,2)	19,902 (0,2)
Liver	841,080 (4,7)	781,631 (8,2)
Lung	2,093,876 (11,6)	1,761,007 (18,4)
Nasopharyngeal	129,079 (0,7)	72,987 (0,8)
Non-Hodgkin lymphoma	509,590 (2,8)	248,724 (2,6)
Pancreatic	458,918 (2,5)	432,242 (4,5)

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
