# Peer review of "Microbes and Cancer: Friends or Faux?"

_ijms, 2020, doi:10.3390/ijms21093115_

Round 1

Reviewer 1 Report

The manuscript titled “Microbes and cancer: friends or faux?”, by Azevedo et al, is well written and provides extensive information on the involvement and interaction of microbiota and cancer initiation and progression. However, the manuscript is very descriptive and does not logically dissect on the “title point”, whether the microbiota is friends of faux. Although the first three points are discussed well, however, there is a lack of information and key references are missing while discussing the role of tumor microbiota in cancer therapy and efficacy.

The key idea is missing how gut microbiota regulates cancer therapy and efficacy in the case of chemotherapy and immunotherapy. A thorough discussion will be important to cohesively discuss how modulation/ alteration of gut microbiota can change the landscape of cancer initiation and progression; as well as change the efficacy and toxicity of cancer therapy.

I would also suggest building a figure, on different microbiota and how they are impacting different organs while causing cancer or having a protective impact.

Furthermore, a critical discussion is suggested to establish the fact that whether the microbiota is a friend or Faux.

Author Response

Reviewer 1

“The manuscript titled “Microbes and cancer: friends or faux?”, by Azevedo et al, is well written and provides extensive information on the involvement and interaction of microbiota and cancer initiation and progression. However, the manuscript is very descriptive and does not logically dissect on the “title point”, whether the microbiota is friends of faux. Although the first three points are discussed well, however, there is a lack of information and key references are missing while discussing the role of tumor microbiota in cancer therapy and efficacy.

The key idea is missing how gut microbiota regulates cancer therapy and efficacy in the case of chemotherapy and immunotherapy. A thorough discussion will be important to cohesively discuss how modulation/ alteration of gut microbiota can change the landscape of cancer initiation and progression; as well as change the efficacy and toxicity of cancer therapy.”

R: We agree with the reviewer that points 4. Tissue Microbioma and Cancer Susceptibility and 5. Role of Microbioma in Cancer Drug Response and Toxicity should be more complete and include more discussion on the role of tumor microbiota in cancer therapy and efficacy.

Concerning the point 4.1. Breast cancer, we added new information and more up to date references; point 4.2.b) Colorectal cancer, we added new information supporting that intestinal bacteria play an important role in CRC development; point 4.2.c) Pancreatic cancer, we highlighted several risk aspects related to pancreatic cancer that are linked to microbiome modification and established the associated between oral microbiome with periodontitis and a higher risk of pancreatic cancer. The Association of H. pylori and pancreatic cancer is also debated. In this point, more recent references were also added; point 4.2.d) Oral cancer, also recent studies were introduced.

Regarding point 5. Role of Microbioma in Cancer Drug Response and Toxicity, more information was added on the role of bacteria in response to chemotherapy, but also in the response to the recent immunotherapies, which was lacking, as well summarized information on the response to other therapeutic modalities, namely radiotherapy and surgery. Also, additional studies on chemotherapy toxicity were added.

I would also suggest building a figure, on different microbiota and how they are impacting different organs while causing cancer or having a protective impact.

Furthermore, a critical discussion is suggested to establish the fact that whether the microbiota is a friend or Faux.

R: As suggested, we included a new figure (Figure 2) in the manuscript that summarizes the data on microbiota in normal and cancer organ/tissues. Also, some sentences were added to the Conclusions section to reinforce the idea of the role of microbiota as friends or faux.

Alterations are highlighted in green.

Reviewer 2 Report

"Microbes and cancer: friends or faux?" is an interesting review. I think Authors did a good work. I have only found a little difficult to keep the thread of the reading, because of the numerous paragraphs. Therefore, it could be useful to create some subsection, for example:

1.Introduction

2.Infection and cancer

  2.1 Virus infections...a)...b)....c)....various viruses

  2.2 Bacterial infections...a)...b)....c)....various bacteria

  2.3 Fungal infections

  2.4 Helmitis infections

3. Susceptibility of Cancer Patients to Acquire Infectious Diseases 

  3.1 Bacterial infections...

  3.2 Fungal infections...

4. Tissue Microbioma and Cancer Susceptibility 

  4.1 Breast cancer

  4.2, 4.3... Other cancers

5. Role of Microbioma in Cancer Drug Response 

6. Conclusions

Author Response

Reviewer 2

"Microbes and cancer: friends or faux?" is an interesting review. I think Authors did a good work. I have only found a little difficult to keep the thread of the reading, because of the numerous paragraphs. Therefore, it could be useful to create some subsection, for example:

1.Introduction

2.Infection and cancer

  2.1 Virus infections...a)...b)....c)....various viruses

  2.2 Bacterial infections...a)...b)....c)....various bacteria

  2.3 Fungal infections

  2.4 Helmitis infections

  1. Susceptibility of Cancer Patients to Acquire Infectious Diseases

  3.1 Bacterial infections...

  3.2 Fungal infections...

  1. Tissue Microbioma and Cancer Susceptibility

  4.1 Breast cancer

  4.2, 4.3... Other cancers

  1. Role of Microbioma in Cancer Drug Response
  2. Conclusions…..”

R: As suggested by the reviewer, we created subsections in the text to help to guide the reader throughout the text.

Alterations are highlighted in green.